# Dual Role of *Sitophilus zeamais*: A Maize Storage Pest and a Potential Edible Protein Source

**DOI:** 10.3390/insects16050531

**Published:** 2025-05-16

**Authors:** Soledad Mora Vásquez, Silverio García-Lara

**Affiliations:** Tecnologico de Monterrey, Escuela de Ingeniería y Ciencias, Ave. Eugenio Garza Sada 2501, Monterrey 64849, Mexico

**Keywords:** insects, *Sitophilus zeamais*, protein, amino acid profile, food safety, nutritional assessment

## Abstract

Maize experiences significant post-harvest losses due to infestations by *Sitophilus zeamais*. This study investigates the potential of *S. zeamais* as a protein source. The weevils were processed into flour and evaluated for food safety, protein content, and amino acid profile. The resulting flour met safety standards, contained 48.1% protein, and was rich in isoleucine, valine, and threonine, although it lacked some essential amino acids. Despite these limitations, *S. zeamais* flour could serve as a viable protein source for both food and feed applications. Incorporating *S. zeamais* flour into food and feed systems could contribute to improved food security.

## 1. Introduction

Maize post-harvest losses due to insect infestations pose a major challenge to food security, particularly in maize-dependent regions where maize serves as an important staple crop, providing primary nutrition for human populations and serving as a key component of livestock feed. Exploring alternative uses for insect pests, such as *Sitophilus zeamais*, could help mitigate these losses while contributing to sustainable protein sources for both food and feed applications. Among the most destructive stored-product pests is *Sitophilus zeamais* Motschulsky (Coleoptera: Curculionidae), commonly known as the maize weevil—an insect species that significantly reduces grain weight, depletes nutritional quality, and impairs germination potential, leading to substantial economic losses [1]. *S. zeamais* thrives in warm and humid environments, with females capable of laying up to 575 eggs within maize kernels, where the larvae develop internally, consuming the endosperm and reducing grain integrity. Infestations are widespread across tropical and subtropical regions, including Asia, Africa, and the Americas, where post-harvest storage systems remain vulnerable to weevil proliferation [2].

Traditional pest management strategies for *S. zeamais* include chemical fumigation, hermetic storage, and biological control methods, such as the use of parasitoid wasps (*Anisopteromalus calandrae*) and entomopathogenic fungi (*Beauveria bassiana*) [3,4]. The growing challenges associated with insecticide resistance, environmental degradation, and pesticide residue accumulation have intensified the search for alternative strategies within integrated pest management frameworks [5]. Given the increasing interest in alternative and sustainable protein sources [6], researchers are increasingly exploring new edible insect species [7] that may offer innovative opportunities to address both food system vulnerabilities and pest control. In this context, the potential utilization of *Sitophilus zeamais* as an edible insect remains largely unexplored. This approach could provide dual benefits of reducing post-harvest maize losses through targeted pest harvesting while also contributing to food security by supplying a sustainable source of protein and essential nutrients.

Insect consumption (entomophagy) is a widespread practice in many cultures, with more than 1900 insect species recognized as edible [8]. Various stored-product pests have historically been consumed, including locusts (*Schistocerca gregaria*), palm weevils (*Rhynchophorus* spp.), and termites (*Macrotermes bellicosus*), demonstrating that the harvesting of pest species for animal or human consumption can serve as a sustainable food production strategy while reducing agricultural losses [9,10,11,12]. In the case of *S. zeamais*, previous studies have documented its consumption in Ghana [13], Nigeria [14], and the Philippines [15]. These reports highlight its chemical composition and mineral content, yet its full nutritional profile, particularly its protein and amino acid composition, remains underexplored.

This study aims to evaluate the nutritional profile and edibility of *Sitophilus zeamais*, focusing on its potential as a sustainable protein source while considering its implications for pest control strategies in maize storage systems. By assessing its protein content, amino acid composition, and microbiological safety, this research contributes to the broader discourse on stored-product pest management, maize post-harvest preservation, and alternative protein sources. Recognizing the emerging potential of *S. zeamais* as a food source could create new opportunities for integrating sustainable pest management strategies while contributing to food security solutions.

## 2. Materials and Methods

### 2.1. Insect Pest Culture

The cultivation of *Sitophilus zeamais* was carried out in the Postharvest Biotechnology Laboratory at Tecnológico de Monterrey, Mexico. Adult specimens of *S. zeamais* were collected from stored maize in Agua Fría, Mexico, and cultured on white maize (single-cross dent hybrid) for four generational cycles under controlled conditions: 27 ± 1 °C, 70 ± 5% relative humidity (RH), and a 12:12 h light/dark photoperiod [16]. Insects were reared in 16 oz (473 mL) Regular Mouth Mason jars (approximately 12.7 cm in height and 7.6 cm in diameter) with airtight lids and bands (Ball^®^, Newell Brands Inc., Atlanta, GA, USA)**.** After two months of cultivation, adult insects were collected, washed with distilled water, surface-disinfected using 90% ethanol for 2 min, and subsequently dried. The dried insects were ground into a fine powder using a cyclone mill equipped with a 1 mm screen to ensure uniformity of the sample. The resulting powdered insect material was used in subsequent biochemical and microbiological analyses.

### 2.2. Food Safety Analysis

A microbiological assay for food safety was carried out following the guidelines stated by the standard methods (Mexican Official Norms NOM-122-SSA1-1994 [17]). For the analysis, 10 g of insect powder was weighed into sterile containers and diluted with 90 mL of sterile diluent. Prior to analysis, the frozen sample was thawed under refrigeration (4–8 °C) for 18 to 24 h, following standard microbiological preparation procedures [18]. Samples were analyzed for aerobic mesophilic bacteria, fungi, yeast, total coliforms, *Escherichia coli*, *Staphylococcus aureus*, and *Salmonella* spp. Aerobic mesophilic bacteria were cultured on Plate Count Agar [18], fungi and yeast on Potato Dextrose Agar [19], and total coliforms and *E. coli* on Brilliant Green Bile Broth [20]. *Staphylococcus aureus* was identified using Baird-Parker medium [21], and *Salmonella* spp. on Xylose Lysine Deoxycholate (XLD) or Hektoen Enteric Agar [22]. All culture media and microbiological reagents were purchased from Sigma-Aldrich (St. Louis, MO, USA), unless otherwise stated. All inoculated plates were incubated under conditions based on standard procedures, including temperature, time, and atmosphere appropriate to the target microorganism [17]. After incubation, microbial colonies were enumerated and key morphological traits—such as size, shape, color, and texture—were assessed in accordance with standard microbiological protocols. All analyses were performed in triplicate.

### 2.3. Extraction of Salt-Soluble Proteins

Salt-soluble proteins were extracted from the sample following the method described by Kim et al. [23]. Specifically, 9 mL of a 0.5 M saline solution was added to 1.5 g of finely ground insect sample in a test tube containing 3 g of glass beads. The test tube was then placed in a shaker and incubated at 4 °C for 2 h. After incubation, the sample was centrifuged at 10,000× *g* for 20 min. A second extraction was performed on the resulting pellet by adding an additional 9 mL of saline solution, shaking the mixture at 4 °C for 1 h, and centrifuging under the same conditions. The supernatants obtained from both extraction steps were combined to yield the salt-soluble protein fraction of the insect sample [23]. Our experiments, conducted in triplicate, confirmed that water-soluble proteins were negligible, thereby justifying the exclusive focus on salt-soluble proteins.

### 2.4. Protein Quantification

Total and salt-soluble protein content was determined using the Kjeldahl method (AOAC Method 928.08), applying a nitrogen-to-protein conversion factor of 5.30 instead of the conventional 6.25. This adjustment was made to mitigate protein overestimation, as the insect cuticle contains substantial amounts of fibrous chitin along with proteins that are tightly embedded within its matrix [24]. A total of 0.1 g of the sample was placed into a digestion flask containing 0.05 g of CuSO_4_ and 1.95 g of K_2_SO_4_, and subsequently 3 mL of H_2_SO_4_ was added. The mixture was digested on a heating grill for 1 h. After digestion, the mixture was diluted with 10 mL of distilled water, followed by the addition of 10 mL of 50% NaOH. The resulting solution was then distilled into a receiver containing an indicator solution, and titration was carried out with 0.200 N HCl until the sample turned transparent [25].

### 2.5. Protein Quality

Approximately 500 mg of *Sitophilus zeamais* flour was accurately weighed and subjected to hydrolysis. For the stable amino acids, isoleucine, leucine, lysine, phenylalanine, threonine, valine, and tyrosine, samples (approximately 500 mg) were hydrolyzed in 6 N HCl containing 0.1% phenol at 110 °C for 24 h under a nitrogen atmosphere. In contrast, tryptophan, which is labile under acidic conditions, was hydrolyzed using 4 M NaOH at 110 °C for 16–18 h, followed by neutralization prior to analysis. After hydrolysis, the sample was cooled to room temperature and the hydrolysate was filtered through a 0.45 µm membrane filter. The filtrate was subsequently evaporated to dryness under reduced pressure at 40 °C and reconstituted in 5 mL of mobile phase. The reconstituted sample was analyzed using high-performance liquid chromatography (HPLC) coupled with an evaporative light scattering detector (ELSD; Agilent Technologies, Santa Clara, CA, USA). Separation was achieved on a reversed-phase C18 column (4.6 mm × 250 mm, 5 µm particle size) maintained at 40 °C. The mobile phases consisted of (A) 0.1% trifluoroacetic acid (TFA) in water and (B) acetonitrile. A gradient elution program was employed, starting at 95% A and 5% B, with a gradual increase in the proportion of B over a 30 min run time to achieve optimal separation of individual amino acids. The flow rate was maintained at 1.0 mL/min, and the injection volume was set to 20 µL. The ELSD was operated under the following conditions: nebulizer gas (nitrogen) flow rate was set to 2.5 L/min, the drift tube temperature was maintained at 90 °C, and the detector gain was optimized to ensure maximum sensitivity for the analytes of interest. Calibration curves were generated using standard solutions of individual amino acids prepared in the same mobile phase, covering a range of concentrations to ensure accurate quantification [26]. The experiment was performed in duplicate. The obtained results were subsequently analyzed by comparing them with the amino acid requirements for infants during the growth stage, as specified by the Food and Agriculture Organization [27].

### 2.6. Statistical Analysis

All primary parameters were expressed as means ± standard deviations. Statistical analyses were conducted using analysis of variance (ANOVA) in the Minitab 19 statistical software (Minitab Inc., State College, PA, USA). 

## 3. Results

### 3.1. Food Safety Analysis

The microbiological analysis of Sitophilus zeamais flour (Table 1) showed aerobic mesophilic bacteria at 590 UFC/g, well below the maximum limit of 100,000 UFC/g. Fungi, yeast, and total coliforms were not detected (<10 UFC/g), and Staphylococcus aureus was present at <10 UFC/g, within the allowable limit of 100 UFC/g. *Salmonella* spp. and Escherichia coli were absent.

### 3.2. Protein Quantification

The protein content of *Sitophilus zeamais* flour was 48.1 ± 0.3% on a dry matter basis, with a salt-soluble protein fraction of 6.6 ± 1.3%, representing approximately 13.7% of the total protein (Table 2).

### 3.3. Protein Quality and Amino Acid Composition

The amino acid profile of Sitophilus zeamais raw flour (Table 3) shows that isoleucine, valine, and threonine exceeded FAO reference values by 40%, 65%, and 24%, respectively. Aromatic amino acids (phenylalanine and tyrosine) were present at more than double the recommended concentration, with an amino acid score of 2.0. In contrast, leucine, lysine, sulfur amino acids (methionine and cysteine), and tryptophan had lower amino acid scores, ranging from 0.7 to 0.9.

## 4. Discussion

### 4.1. Food Safety Analysis

The microbiological analysis of *Sitophilus zeamais* flour indicates that the product complies with established food safety standards. Aerobic mesophilic bacteria were present at 590 UFC/g, a value significantly below the maximum allowable limit of 100,000 UFC/g as defined by NOM-122-SSA1-1994 [17]. Moreover, the levels of fungi and yeast were below the detection threshold (<10 UFC/g), and total coliform bacteria were not detected. *Staphylococcus aureus* was identified at <10 UFC/g, which is well within the permissible limit of 100 UFC/g. Notably, *Salmonella* spp. and *Escherichia coli* were absent from the sample, further substantiating the microbiological safety of the flour.

Previous research has demonstrated that microbial loads in edible insect products are influenced by multiple factors, including the insects’ inherent microbial content, the impact of processing on bacterial populations, and the risk of secondary contamination [28]. When subjected to appropriate processing techniques, insect-derived flours and other products exhibit microbial profiles that meet or exceed food safety standards. These consistent outcomes across various studies suggest that the application of standardized hygienic practices and controlled processing conditions is crucial for ensuring the safety of edible insect products [29].

From a pest control perspective, these findings are relevant because they indicate that harvesting *S. zeamais* from maize storage facilities for consumption does not introduce additional food safety concerns. If integrated into post-harvest pest management strategies, targeted collection efforts could help reduce infestation rates in stored grains, providing an alternative pest mitigation approach while ensuring nutritional benefits [30].

### 4.2. Protein Quantification

The high total protein content confirms that *S. zeamais* flour is a protein-rich material, a characteristic that is frequently reported in studies focusing on edible insects. However, the salt-soluble fraction, which constitutes approximately 13.7% of the total protein content, represents only a minor component of the overall protein profile. This observation is significant, as salt-soluble proteins generally include those involved in enzymatic functions and other cellular activities that require ionic interactions for stability and solubility. In contrast, the bulk of the protein content may be composed of proteins with different solubility properties, such as water-insoluble or structural proteins.

The relatively low proportion of salt-soluble proteins could be attributed to the biological characteristics and functional roles of proteins within *S. zeamais*. It is conceivable that the majority of the proteins are either bound to cellular structures or exist in forms that do not readily solubilize in saline solutions. In studies involving *Tenebrio molitor*, the salt-soluble protein fraction has been shown to be significantly more digestible compared to the insoluble fraction. This soluble fraction was notably enriched in hemolymph proteins and enzymes such as alpha-amylase, which play essential roles in nutrient transport and carbohydrate metabolism [31]. These findings suggest that the high digestibility of the soluble proteins is largely attributable to their specific composition, which favors proteins involved in physiological functions over more structurally bound muscle proteins. This distinction in protein solubility has implications for both the biological understanding of the insect and potential control strategies. For instance, detailed characterization of the protein profile could provide insights into metabolic pathways critical for insect survival, thereby identifying novel targets for pest management interventions.

Moreover, the substantial overall protein content underscores the potential for utilizing *S. zeamais* as a source of protein in various applications, including animal feed and human food products, provided that safety and processing standards are met. The identification of specific protein fractions, such as the salt-soluble fraction, may also aid in the development of extraction techniques that maximize yield and functional quality, ultimately contributing to the valorization of insect biomass in sustainable food systems.

Numerous studies have quantified the protein content in various insect families, reporting high overall values: Saturniidae (40–50%), Notodontidae (42–45%), Gryllidae (53%), Acrididae (76%), and Tenebrionidae (52%) [31,32,33,34]. *Sitophilus zeamais* flour exhibits a total protein content of 48.1 ± 0.3%. This value falls within the range observed for other insect species, underscoring the potential of *S. zeamais* as a protein-rich resource for nutritional applications.

In conclusion, the protein profile of *S. zeamais* flour, characterized by a high total protein content and a modest salt-soluble fraction, offers valuable insights into the insect’s biology. These findings not only enhance our understanding of protein composition in corn insect pests but also pave the way for future research aimed at exploiting these biological resources for innovative control and utilization technologies.

### 4.3. Protein Quality and Amino Acid Composition

The amino acid profile of *Sitophilus zeamais* raw flour, as detailed in Table 3, reveals a complex pattern of nutritional adequacy and limitation when compared to the FAO-recommended amino acid requirements for infants in the growing stage [27]. Notably, the levels of isoleucine, valine, and threonine exceed the reference values by 40%, 65%, and 24%, respectively, and the aromatic amino acids (combined phenylalanine and tyrosine) are present at more than double the recommended concentration, yielding an amino acid score of 2.0. Conversely, leucine, lysine, sulfur amino acids (methionine and cysteine combined), and tryptophan fall below the reference levels, with amino acid scores ranging from 0.7 to 0.9, thereby identifying them as limiting factors in the protein quality of the flour. This duality in amino acid composition suggests that while *S. zeamais* flour could serve as a valuable protein source, particularly in applications requiring high levels of certain essential amino acids, its deficiencies in others may necessitate formulation adjustments or supplementation to achieve a balanced nutritional profile.

These results align with previous studies on edible insects, which have similarly reported high protein content with favorable levels of certain essential amino acids, though often with one or more limiting amino acids that restrict the flour’s use as a complete protein source [33]. Therefore, while *S. zeamais* raw flour exhibits promising nutritional attributes, its integration into food formulations may require strategies to overcome its amino acid limitations. These could include fortification with sulfur-containing amino acids, such as cysteine or methionine, or blending with complementary protein sources—for example, legumes (e.g., soy or lentils), eggs, or dairy proteins—that provide the deficient amino acids [35]. Such approaches are commonly used in food product development to improve amino acid balance and enhance the overall nutritional quality of novel protein ingredients.

### 4.4. Pest Control and Sustainable Utilization Strategies

Several precedents have demonstrated the potential of utilizing agricultural pests as food sources to mitigate their negative impacts. For instance, locust harvesting—specifically of *Schistocerca gregaria* and *Locusta migratoria*—has contributed to reducing swarm sizes in Africa while simultaneously providing a high-protein food source [36]. Similarly, the collection of palm weevil larvae (*Rhynchophorus* spp.) has been shown to lower infestation rates in oil palm plantations, with these larvae consumed as a delicacy in West Africa and Southeast Asia [37]. Additionally, grasshopper collection in Uganda has proven effective in managing outbreaks of *Ruspolia differens* [38]. These examples suggest that harvesting pests for consumption can serve as a viable control strategy. In light of these findings, further research should evaluate the feasibility of implementing controlled *Sitophilus zeamais* collection programs. Such programs could reduce post-harvest losses in maize storage facilities, provide a locally available protein source in regions facing food insecurity, and diminish pesticide dependency by incorporating entomophagy into pest management strategies. Moreover, assessing consumer acceptance and examining regulatory frameworks will be essential to ensure the scalability and economic viability of this integrated approach.

## 5. Conclusions

This study highlights the dual role of *Sitophilus zeamais* as both an agricultural pest and a potential alternative protein source. The high protein content and essential amino acid composition of *S. zeamais* support its nutritional viability, while its presence in maize storage systems suggests an opportunity for sustainable pest control through targeted harvesting. By integrating biological control methods with entomophagy, it may be possible to mitigate post-harvest losses while contributing to food security and sustainable agriculture.

While the present study provides an initial characterization of *S. zeamais* from a nutritional and microbiological perspective, it should be regarded as a preliminary step toward broader evaluations. A comprehensive assessment of its safety for human consumption—including allergenic potential, toxicological risks, and long-term effects—remains essential. Accordingly, future research should address these dimensions, in addition to exploring scalability, consumer acceptance, and regulatory frameworks, to determine the feasibility of incorporating *S. zeamais* into integrated pest management and sustainable food systems.

## Figures and Tables

**Table 1 insects-16-00531-t001:** Microbiological test for food safety analysis of *Sitophilus zeamais* flour.

Microorganism	Result (UFC/g) **	Maximum Limit (UFC/g) *
Aerobic mesophilic bacteria	590	100,000
Fungi	<10	<10
Yeast	<10	<10
Total coliform	None	100
*Staphylococcus aureus*	<10	100
*Salmonella* spp.	None	Negative
*Escherichia coli*	None	Negative

* In accordance with the Health Secretary of Mexico in order to guarantee the quality of food safety analysis (NOM-122-SSA1-1994). ** Data represent the mean of three independent replicates.

**Table 2 insects-16-00531-t002:** Total protein and protein profile of *Sitophilus zeamais* flour.

Protein	Content (%) *
Salt-soluble fraction	6.6 ± 1.3
Total Content	48.1 ± 0.3

* Mean values ± standard deviation (n = 3).

**Table 3 insects-16-00531-t003:** Amino acid profile and protein quality of *Sitophilus zeamais* raw flour.

Amino Acid	Reference (mg/g Protein) *	*S. zeamais* (mg/g Protein) **	Difference	Amino Acid Score ***
Isoleucine	30	42	+12	1.4
Leucine	61	55.4	−5.6	0.9
Lysine	48	40.6	−7.4	0.8
Methionine	-	10.9	-	
Cysteine	-	5.6	-	
Sulfur AA (Met + Cys)	23	16.5	−6.5	0.7
Phenylalanine	-	30.5	-	
Tyrosine	-	53.3	-	
Aromatic AA (Phe + Tyr)	41	83.8	+43	2.0
Tryptophan	6.6	4.8	−1.8	0.7
Valine	40	66.3	+24	1.7
Threonine	25	30.9	+5.9	1.2

* Reference based on recommended AA (amino acids) for an infant at growing stage [27]. ** Data correspond to the average of two determinations with an analytical variability of less than 1%. *** *Amino acid score =*
$\frac{mgAA\;in\;1\;g\;of\;S.zeamais\;protein}{mg\;AA\;in\;1\;g\;of\;reference\;protein}.$

## Data Availability

The data presented in this study are available on request from the corresponding author due to confidentiality agreements with collaborating institutions. Graphic Abstract created in BioRender, https://BioRender.com/gexem4m (accessed on 6 March 2025).

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
