# Peer review of "Dual Role of Sitophilus zeamais: A Maize Storage Pest and a Potential Edible Protein Source"

_insects, 2025, doi:10.3390/insects16050531_

Round 1
Reviewer 1 Report
Comments and Suggestions for Authors
In this manuscript, the authors investigates the potential of S. zeamais as a protein source for human, and they detected the microbiological safety, protein content, and amino acid composition in S. zeamais. The manuscript was well written and easy to understand, but the contents are not enough as safety is extremely important.
- The authors collected S. zeamais and reared in lab for 4 generations, then using them as samples, but I wonder what about the safety of the original insects.
- In table 1. I think the authors should provide the exact numbers of different microbiology instead of < 10.
- Although the authors studied the aa content in the protein, but I wonder whether there was any allergen present in the insect?
Author Response
Comment: In this manuscript, the authors investigates the potential of S. zeamais as a protein source for human, and they detected the microbiological safety, protein content, and amino acid composition in S. zeamais. The manuscript was well written and easy to understand, but the contents are not enough as safety is extremely important.
Response:
We sincerely thank the reviewer for their thoughtful assessment and we fully agree that safety is of paramount importance when evaluating novel protein sources for human consumption. In this initial study, our aim was to provide a foundational characterization of S. zeamais, focusing on microbiological safety, protein content, and amino acid composition, as a preliminary step toward assessing its potential as a nutritional ingredient. We recognize that a more comprehensive safety evaluation would ideally include analyses of allergenic potential, toxicological profiling, and long-term exposure studies. While these aspects were beyond the scope of the present work, we have now emphasized in the revised manuscript that our findings represent a starting point, and that further research is needed to fully elucidate the safety profile of S. zeamais for human consumption. We have also included a statement highlighting the importance of future studies addressing these critical aspects (see conclusion). We are grateful for this valuable suggestion, which has helped us to improve the contextual framing of our work within the broader safety considerations required for novel food sources.
Comment 1: The authors collected S. zeamais and reared in lab for 4 generations, then using them as samples, but I wonder what about the safety of the original insects.
Response:
To address potential concerns regarding the safety and origin of the insects, we clarify that S. zeamais specimens were collected from commercial maize storage silos. These facilities handle maize intended for industrial purposes.
To further mitigate any potential environmental or biological contamination, the insects were maintained under controlled laboratory conditions and reared for four successive generations prior to sample collection. This rearing period was intended not only to stabilize the population but also to minimize the influence of external variables, ensuring a healthy and standardized colony. We believe this approach provides a reasonable safeguard while enabling a reproducible characterization of the insect material.
Comment 2: In table 1. I think the authors should provide the exact numbers of different microbiology instead of < 10.
We sincerely thank the reviewer for this observation. We would like to clarify that the notation "<10 CFU/g" reflects the methodological detection limit of the microbiological analysis performed. In standard practice, due to the nature of dilution techniques and the sample volumes tested, the lowest reliably reportable count is typically 10 CFU/g. When no colonies are observed on the plated media, this does not necessarily indicate a complete absence of microorganisms; rather, it means that if any microorganisms are present, they exist at levels below the detection threshold of the method employed.
Reporting "0 CFU/g" could suggest an absolute absence, which cannot be definitively confirmed without the use of much larger sample volumes or more sensitive detection techniques. Therefore, to maintain scientific accuracy and transparency, results are conventionally reported as "<10 CFU/g" in such cases. This approach is widely adopted in microbiological analyses to avoid overinterpretation of negative results.
We hope this explanation clarifies our reporting choice, and we remain fully open to incorporating any further adjustments the reviewer may deem necessary.
Comment 3: Although the authors studied the aa content in the protein, but I wonder whether there was any allergen present in the insect?
This is an important point, and we thank the reviewer for raising it. While we acknowledge the possibility of allergenic proteins in insect-derived materials, our study focused specifically on the soluble protein fraction. This choice was made intentionally, as many known insect allergens—such as tropomyosin and arginine kinase—are typically associated with the exoskeletal components, which are largely absent from the soluble fraction. To the best of our knowledge, no confirmed allergenic proteins have been identified in S. zeamais to date; however, we agree that further research is essential to comprehensively assess allergenicity. We now include a sentence acknowledging the need for future studies to evaluate potential allergenic risks in S. zeamais protein extracts (see final conclusion).
Reviewer 2 Report
Comments and Suggestions for Authors
Congratulations to the authors for their interesting and innovative work. The manuscript is well-written and presents a novel approach by suggesting a new insect species as a potential protein source. However, some aspects require further improvement and clarification. In particular, the Materials and Methods section should be more detailed to ensure the reproducibility of the study.
Abstract: Line 17: Please add italics to Zea mays.
Introduction:
Line 25: Please add italics to Salmonella and Escherichia coli, and add "sp." after Salmonella.
Line 32: Could you specify what means "enhanced food"? The sentence appears to be incomplete.
Line 40: Please add italics to Sitophilus zeamais.
Line 43: Please include the taxonomic authority, specifying the order and family of the species.
Line 55: The transition from the problem to the potential use of S. zeamais as an edible insect could be more fluid. For example: "Given the growing interest in alternative and sustainable protein sources (Citation), researchers are increasingly exploring new edible insect species (Citation) that may provide an innovative opportunity to simultaneously address post-harvest losses and the rising demand for protein."
Line 59: Please remove the italics from "entomophagy".
Materials and Methods: Could you specify how many replications were performed for each experiment?
Line 81: Please specify the type of container/cage used for rearing the species, including dimensions.
Line 87: How much powder was used for the analysis? Please specify the amount.
Line 99: Could you clarify what the "appropriate conditions for microbial growth" are?
Line 100: Please specify the "morphological characteristics" analyzed.
Line 103: Could you justify the choice of this extraction method over other available methods?
Line 106: Please add a space before "°C" and ensure consistency throughout the manuscript.
Results:
Lines 161, 163, 164: Please add italics to the species names (Sitophilus zeamais, Staphylococcus aureus, Salmonella sp., Escherichia coli).
Line 183: Please add italics to the species name.
Discussion:
Line 281: Could you suggest ways to overcome the identified limitations? For example, would it be possible to integrate S. zeamais flour with other protein sources to balance its nutritional profile?
Author Response
Please note that our responses appear in black color font immediately following each reviewer comment, which is shown in gray, to facilitate a clear and organized review process.
Congratulations to the authors for their interesting and innovative work. The manuscript is well-written and presents a novel approach by suggesting a new insect species as a potential protein source. However, some aspects require further improvement and clarification. In particular, the Materials and Methods section should be more detailed to ensure the reproducibility of the study.
Thank you very much for the encouraging feedback and constructive suggestions. In response, we have expanded the Materials and Methods section with additional details to enhance clarity and ensure reproducibility.
Abstract: Line 17: Please add italics to Zea mays. Zea mays has been italicized as requested.
Introduction:
Line 25: Please add italics to Salmonella and Escherichia coli, and add "sp." after Salmonella. Salmonella and Escherichia coli have been italicized, and "sp." has been added after Salmonella as requested.
Line 32: Could you specify what means "enhanced food"? The sentence appears to be incomplete.
We agree that the term "enhanced food" was vague and that the sentence required clarification. The sentence has been revised to better reflect the intended meaning:
“These findings suggest that, despite certain limitations, S. zeamais flour represents a viable protein source. Integrating targeted insect harvesting for protein into pest management strategies could help reduce post-harvest losses and contribute to improved food security and nutritional availability.”
Line 40: Please add italics to Sitophilus zeamais. Sitophilus zeamais has been italicized as requested and we have reviewed the entire manuscript to ensure consistent formatting
Line 43: Please include the taxonomic authority, specifying the order and family of the species. We appreciate the reviewer’s suggestion and have updated the sentence to include the taxonomic authority, order, and family of Sitophilus zeamais.
“Among the most destructive stored-product pests is Sitophilus zeamais Motschulsky (Coleoptera: Curculionidae), commonly known as the maize weevil, an insect species that significantly reduces grain weight, depletes nutritional quality, and impairs germination potential, leading to substantial economic losses.”
Line 55: The transition from the problem to the potential use of S. zeamais as an edible insect could be more fluid. For example: "Given the growing interest in alternative and sustainable protein sources (Citation), researchers are increasingly exploring new edible insect species (Citation) that may provide an innovative opportunity to simultaneously address post-harvest losses and the rising demand for protein."
We agree that improving the transition between the discussion of pest-related challenges and the proposal of S. zeamais as a potential edible insect enhances the clarity and flow of the introduction. Accordingly, we have revised the paragraph to incorporate this framing and to better reflect the broader context of sustainable protein research. The revised version now reads:
"The growing challenges associated with insecticide resistance, environmental degradation, and pesticide residue accumulation have intensified the search for alternative strategies within integrated pest management frameworks [5]. Given the increasing interest in alternative and sustainable protein sources [6], researchers are increasingly exploring new edible insect species [7] that may offer innovative opportunities to address both food system vulnerabilities and pest control. In this context, the potential utilization of Sitophilus zeamais as an edible insect remains largely unexplored. This approach could provide dual benefits: reducing post-harvest maize losses through targeted pest harvesting, while also contributing to food security by supplying a sustainable source of protein and essential nutrients."
Line 59: Please remove the italics from "entomophagy". The italics have been removed from "entomophagy" as requested.
Materials and Methods: Could you specify how many replications were performed for each experiment?
Thank you very much for your helpful comment regarding the number of replications performed in each experiment. We have now clarified this information in the manuscript as follows:
- In the food safety analysis methods, we added: “All analyses were performed in triplicate.”
- In the extraction of salt-soluble proteins section, we included: “Our experiments, conducted in triplicate, confirmed that water-soluble proteins were negligible, thereby justifying the exclusive focus on salt-soluble proteins.”
- In the protein quality analysis section, we specified: “Calibration curves were generated using standard solutions of individual amino acids prepared in the same mobile phase, covering a range of concentrations to ensure accurate quantification [26]. The experiment was performed in duplicate. The obtained results were subsequently analyzed by comparing them with the amino acid requirements for infants during the growth stage, as specified by the Food and Agriculture Organization [27].”
Line 81: Please specify the type of container/cage used for rearing the species, including dimensions.
Thank you for your suggestion. We have added the following information to the Methods section for clarification:
“The insects were reared in 16 oz (473 mL) Regular Mouth Mason jars (approximately 12.7 cm in height and 7.6 cm in diameter) with airtight lids and bands, manufactured in the USA.”
Line 87: How much powder was used for the analysis? Please specify the amount.
In each analysis, the specific amount of sample used is indicated in the corresponding section. However, the amount used for microbiological analysis was not previously specified. To address this, we have added the following paragraph to the Methods section:
“For microbiological analysis, 10 grams of insect powder were weighed into sterile containers and diluted with 90 mL of sterile diluent. Prior to analysis, the frozen sample was thawed under refrigeration (4–8 °C) for 18 to 24 hours, following standard microbiological preparation procedures [15].”
Line 99: Could you clarify what the "appropriate conditions for microbial growth" are?
We have revised the text to specify that microbial cultures were incubated under conditions based on standard procedures:
“All inoculated plates were incubated under conditions based on standard procedures, including temperature, time, and atmosphere appropriate to the target microorganism [15]”.
Line 100: Please specify the "morphological characteristics" analyzed.
We have clarified this point in the Methods section by specifying the morphological characteristics analyzed. The revised text now reads:
“After incubation, microbial colonies were enumerated, and key morphological traits—such as size, shape, color, and texture—were assessed in accordance with standard microbiological protocols.”
Line 103: Could you justify the choice of this extraction method over other available methods?
The extraction method we employed, based on Kim (2019), was selected due to its proven efficiency in isolating salt-soluble proteins from insect matrices, particularly in studies aiming to characterize nutritional potential. Compared to other extraction approaches (e.g., water, alkaline, or enzymatic methods), the use of a 0.5 M saline solution provides a milder, non-denaturing environment that preserves protein structure and functionality, which is essential for downstream nutritional and biochemical analyses.
Line 106: Please add a space before "°C" and ensure consistency throughout the manuscript.
Thank you for pointing this out. A space has been added before “°C” at the specified location, and consistency has been ensured throughout the entire manuscript. We appreciate your attention to detail.
Results:
Lines 161, 163, 164: Please add italics to the species names (Sitophilus zeamais, Staphylococcus aureus, Salmonella sp., Escherichia coli).
Italics have been applied to all species names mentioned in Lines 161, 163, and 164, and we have reviewed the entire manuscript to ensure consistent formatting of scientific names throughout.
Line 183: Please add italics to the species name.
As above
Discussion:
Line 281: Could you suggest ways to overcome the identified limitations? For example, would it be possible to integrate S. zeamais flour with other protein sources to balance its nutritional profile?
Thank you for this valuable suggestion. We have revised the discussion to more explicitly address potential strategies to overcome the identified amino acid limitations of S. zeamais flour. Now it reads:
“Therefore, while S. zeamais raw flour exhibits promising nutritional attributes, its integration into food formulations may require strategies to overcome its amino acid limitations. These could include fortification with sulfur-containing amino acids, such as cysteine or methionine, or blending with complementary protein sources—for example, legumes (e.g., soy or lentils), eggs, or dairy proteins—that provide the deficient amino acids [35]. Such approaches are commonly used in food product development to improve amino acid balance and enhance the overall nutritional quality of novel protein ingredients.”